# Cost of Extending the Farm Accountancy Data Network to the Farm Sustainability Data Network: Empirical Evidence

**Hans Vrolijk \*** and **Krijn Poppe**

Wageningen Economic Research, Wageningen University and Research, 6708 PB Wageningen, The Netherlands; kjpoppe@hccnet.nl
\* Correspondence: hans.vrolijk@wur.nl

**Abstract:** The European Green Deal, its Farm to Fork strategy and Biodiversity strategy will set the scene for the future revisions of the Common Agricultural Policy (CAP). The CAP will address an increasing set of objectives, including contributing to the Sustainable Development Goals and the Paris climate agreement. To enable evidence-based policy making and monitoring, the Farm to Fork strategy proposes to extend the current monitoring system to include a broader range of sustainability issues. The current monitoring system called Farm Accountancy Data Network (FADN) has a strong focus on financial and economic data. The FADN is an instrument for monitoring and evaluation of the EU Common Agricultural Policy and collects bookkeeping results from 80,000 farms. The extension to a Farm Sustainability Data Network (FSDN) should include a broader set of indicators on the sustainability performance of farms. This paper estimates the costs of collecting this broader set of sustainability indicators in the FSDN based on the experiences of a pilot in 9 member states and a survey among all member states. The results show that collecting the sustainability data from all farms included in FADN would increase the costs by about 40%. The results show large differences between countries depending on the current costs of data collection and the expected additional work to include sustainability indicators. Given the pressing need for these data, a scenario was developed where sustainability data are collected from a subsample of 15,000 farms. This can be achieved within current budget limits if the current FADN sample would be reduced from 85,000 to 75,000 farms. The discussion section addresses some concerns raised on the extension of FADN to FSDN such as: willingness of farmers, administrative burden, economic background of FADN and the quality of the data.

**Keywords:** FSDN; FADN; data collection; sustainability; data needs; cost estimates

## 1. Introduction

The Green Deal is the European Union's plan to make the EU economy more sustainable [1]. The Farm to Fork Strategy is at the heart of the European Green Deal aiming to make food systems fair, healthy and environmentally friendly [2]. The Farm to Fork Strategy sets clear ambitions with respect to the increase in organic farming, and a reduction in the use of pesticides, fertilisers and antibiotics. The future Common Agricultural Policy (CAP) will show a higher level of ambition to mitigate environmental and climate impacts. The policy is set to shift the emphasis from compliance and rules towards results and performance. The shift to more environmental policies, the interplay between environmental and agricultural policies and the shift to results and performances will bring forward new data needs for monitoring and policy evaluation. Changing data needs are not entirely new [3–5], but the needs are more compelling, given the growing emphasis on environmental and climate impacts. Up till now, the Farm Accountancy Data Network (FADN) has been the main monitoring instrument used to evaluate agricultural policies within the EU. FADN collects financial, economic and structural data, including a profit-and-loss account, balance sheet and income statement on 85,000 farms on a yearly basis (hereafter: the FADN

data). FADN is appreciated by different stakeholders because of its broad use of the data. The benefits consist first of all of public benefits: better policy decisions lead in the end to better social outcomes [6]. In addition, there are private benefits: data can be used by farmers and their advisors [7,8] to improve farm performance through, for example, benchmarking [9]. To enable a continued evidence-based policy making and monitoring, the Farm to Fork Strategy proposes to extend the current FADN to a Farm Sustainability Data Network (FSDN) to include a broader set of indicators on the sustainability performance of farms.

The extension to FSDN could benefit from several indicator frameworks that have been developed by a range of international organisations (such as United Nations millennium development goals, Eurostat agri-environmental indicators, European Environment Agency indicators, OECD agri-environmental indicators, FAO indicators of sustainable development). In addition, several research projects have developed indicator sets (IRENA [10], AE Foodprint [11]). Also at the national level are initiatives to measure the sustainability performance of farms [12–14]. Overlooking these initiatives, we conclude that initiatives differ in level of measurement (farm, regional or national level), empirical implementation (some frameworks exist on paper but it is unclear how data should be collected) or are not harmonised across countries.

The FLINT (Farm Level Indicators for New Topics in policy evaluation) project was funded by the European Commission to test the feasibility of collecting sustainability data at the farm level and to illustrate the value of this type of data to improve policy making [15]. The project defined a list of relevant sustainability themes based on emerging policy needs, a literature review and a review of national initiatives to measure sustainability [16,17]. Finally, 31 policy relevant themes were selected (see Figure 1), and each of the themes was translated into a list of specific data items to be collected at the farm level (hereafter: the FLINT data).

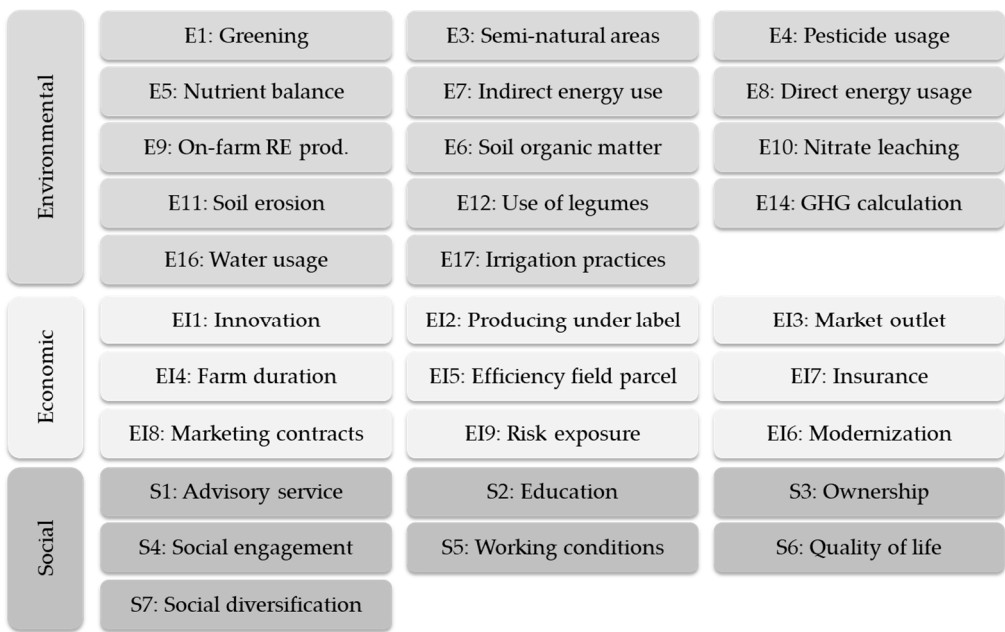

**Figure 1.** Sustainability themes as included in the FLINT data collection.

The feasibility of collecting these data was tested by collecting the defined data items in 9 member states (Ireland, Netherlands, Germany, Poland, Finland, Hungary, Greece, Spain and France) on 1100 farms of different types [18]. The results showed that data collection is possible in the different administrative environments of the national FADN. In general, the FLINT project showed positive experiences of collecting sustainability data and the project showed that farmers are willing to make the data available [19].

The FLINT project has shown how policy analysis benefits from additional data with indicators on the sustainability performance of farms (profit, planet and people aspects). The analyses were used to illustrate how the additional data provide benefits in terms of (1) filling gaps in terms of research methodology (i.e., social performance, economic viability); (2) provide better understanding of the sources of sustainability performance (i.e., impact of land fragmentation, advisory services, age of assets) [20]; (3) provide additional insights into challenges faced by farmers (i.e., trade-offs between environmental and economic performance) [21,22]; and (4) provide more precise recommendations for policy makers (i.e., effect of CAP subsidies on technical efficiency, impact of investment subsidies on age of assets) [23]. These and other applications [24–26] of the FLINT data illustrate the value of sustainability data for research and policy evaluation. Although difficult to quantify, the FLINT data facilitate the pursuit of societal goals, in particular, environmental and those related to rural development, by improving the targeting, efficacy and efficiency of agricultural policies [19]. The project has demonstrated the increase in public value of the FADN data set when data on the sustainability performance of farms are included. Collecting, managing and processing of farm level data in the FADN require large financial resources. Societal benefits come from policy applications [27–29] and research applications [30–33]. An increase in the use of the data by aligning them to the new policy objectives increases the benefits and provides a stronger justification for these expenditures.

Given these developments, it is understandable that the European Commission proposes to create the FSDN and extend the FLINT pilot project to a yearly obligation in all EU member states. Obviously, the decision to adopt this proposal not only involves the benefits as illustrated above, but also the cost of collecting sustainability performance data at the farm level. Until now, little information has been available on the costs of collecting data on the sustainability performance of farms.

The objective of this paper was to estimate the costs of collecting farm data for the new CAP's needs. The results of this paper provide valuable input for further decision making in the European Parliament and Council of Ministers as well as in the implementation of the FSDN in Europe.

## 2. Materials and Methods

Adding the FLINT data to the FADN obviously adds a cost. Estimating these costs is not a trivial issue. We will describe the estimation of the costs of the FLINT data collection in two steps. The first step consists of an estimation of the costs in the 9 pilot countries involved in the FLINT project. In the second step, we make an estimation of the costs for all member states, as the FSDN will become an obligation for all member states. Two separate surveys were used to provide input for these estimations (see Figure 2). The first survey was among data collectors involved in the FLINT data collection and the second survey was among FADN managers from all EU member states.

The survey among data collectors (survey 1 in Figure 2) was done by an online questionnaire. The survey covered the time required for the data collection by type of farming, the incentive scheme used to gain the data from the farmers and a cost estimation. The time required for collecting the data per farm contains information on time needed for preparatory work, farm visit, completion, delivery and control of the data.

The FLINT project was a pilot conducted in 9 countries. To be able to draw conclusions for the EU-28, information on the non-pilot countries was used. A survey was conducted among FADN managers from all EU member states (survey 2 in Figure 2, see Supplementary Materials). The survey aimed to collect data on the organisational setup of the data collection, the scope of the current data collection and the time and budget requirements to collect the current FADN data. All member states responded to the survey. In addition, the results of a study commissioned by the EU Commission on the costs of FADN [6] were used to extrapolate the costs of the pilot countries to all EU member states. With this

approach, we prevent underestimation of the costs in case the (marginal) project costs are relatively low.

| | Hours per farm | Data collection costs per hour |
|---|---|---|
| FADN data (data collection according to current EU monitoring obligations) | Survey 2 (among FADN managers) (missing values based on Bradley and Hill [6]) | Survey 2 (among FADN managers) (missing values based on Bradley and Hill [6]) |
| FLINT data (data collection on additional sustainability indicators) | Survey 1 (among data collectors for pilot countries) Survey 2 (among FADN managers for non-pilot countries) | From survey above |

**Figure 2.** Surveys and external data sources used to estimate hours and costs of data collection.

To crosscheck the data and to be able to impute a value in case of missing values, countries were categorised into three types. The types depended on the extent to which sustainability data are collected in the current situation and whether the data were collected using the same data collection process or a separate process. This approach acknowledges the fact that the increase in costs differ strongly depending on the current processes and costs of data collection. This varies widely due to the different administrative environments: some countries have dedicated staff to collect FADN data, whereas in other countries the data are purchased from accounting offices. The existing data collection processes determine the complexity to collect the new data items. Missing values were imputed by the average of hours belonging to the same type. Outliers were corrected in the same way.

The costs of FADN and FLINT data collection were estimated per country by multiplying the required number of hours for FADN plus FLINT data collection with the data collection costs per hour applicable to a specific country.

The cost estimations per country are subsequently used to estimate the implications for the European Union as a whole. The increase in costs per farm at the EU level is estimated. Two options to maintain the costs at the current level are also described. In the first option, the number of farms in the FADN decreases per member state, based on the estimated costs of collecting FLINT data in that particular country (total costs at the country level remain the same). In the second option, FLINT data are collected from a subsample of farms. The size of this subsample is based on a trade-off between the costs of data collection and the precision of the estimates of the relevant indicators. The relative standard error is calculated based on the FLINT sample and the expected standard error for other sample sizes is calculated (ranging from 5000 to 50,000). The allocation of this sample size to the different countries is based on the optimal or Neymann allocation [34]. The FADN sample size is reduced at the country level to exactly compensate the additional FLINT data collection costs. For these analyses, the FLINT dataset and the FADN dataset were used. Access to the FADN dataset was provided by DG-Agri of the European Commission.

## 3. Results

### 3.1. Estimation of Costs for the FLINT Pilot Countries

During FLINT data collection, consortium partners had to face initial costs (such as training of data collectors, developing and installing IT infrastructure, etc.), which occur only in the first year and will be incorporated into general FADN data collection costs if the European Commission decides to turn the pilot network into an operational EU-wide system.

Based on the responses, there was no special incentive in eight out of nine member states to persuade the farmers to take part in the FLINT project. Where data collection was done by FADN data collectors (Finland, Netherlands, Hungary, Spain), the good relation between data collectors and farmers encouraged participation. Farmers were informed about the aim of the data collection. Only German farms received a financial incentive (150–500 euros per farm) as compensation for the time and effort needed to participate in the FLINT survey.

Figure 3 shows the results of the survey among data collectors who gave an estimation of the time required for collecting the data per farm. On average, 8.5 h per farm were needed, including 3 h for preparatory work, 2.5 h for the farm visit and 3 h for the completion, delivery and control of the data (see Figure 3).

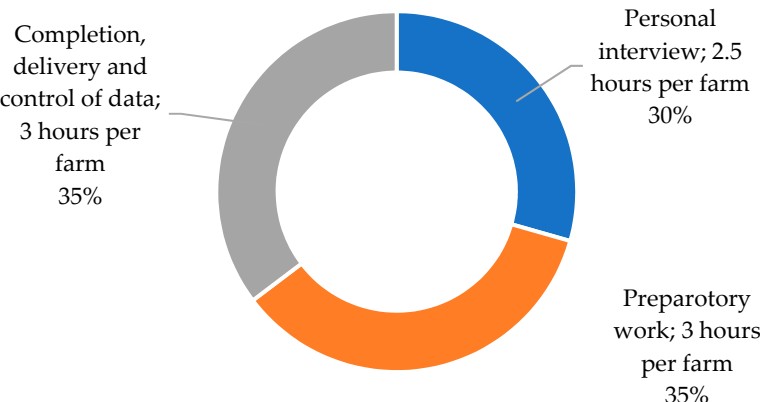

**Figure 3.** Average time required for data collection per farm.

Looking in more detail, a substantial variation among member states concerning the time needed to collect the data can be observed. The results are influenced by the applied data collection methodology, the extent to which FLINT data were already part of the national FADN systems, the number of FLINT indicator themes relevant in the country, as well as the type and the size of a given farm. The average time required for data collection and data processing (including validation) of 8.5 h varies from five to fifteen hours depending on the abovementioned circumstances.

Although the FLINT data collection was based on the FADN data collection methodology, the whole data collection procedure had to be established from the beginning. A new farm return (specification of all data items to be collected at the farm) was implemented, data collectors had to be trained and a new or adapted IT infrastructure was installed for data recording, validation and storage. For those member states (Ireland, Finland, Netherlands, Hungary, Poland) where the FADN data collection is flexible and the system can easily adopt changes, the initial setup costs were relatively low and existing resources were more effectively used.

The total expenditure of data collection is not directly comparable between member states. In Germany, the farmers self-reported their data and they received 150–500 euros per farm, but the data recording into the official FLINT spreadsheet was done by researchers. Poland spent 100 euros per completed questionnaire, but this figure does not contain the cost of data entry. In Spain, Hungary and Finland, on average 300 euros per Farm Return were paid for the data collection (the cost of recording included). The other additional costs, such as recruiting of farms, training of data collectors, validation of data, application of new IT solutions, vary from member state to member state, depending on the administrative environment and existing infrastructure in which FLINT data collection was integrated in the existing data collection process.

### 3.2. Estimation of Costs for All Member States

The results for the estimated costs of collecting FLINT sustainability data are displayed in Table 1. Column 2 classifies the member state on the current level of data collection on sustainability performance indicators and the most likely method to collect the FLINT data, based on indications by FADN committee members for their country. Column 3 gives the number of hours currently needed per farm to collect the FADN data. Column 4 gives the costs for collecting data from one FADN farm (labour costs and in some cases costs for "buying" the data from accounting offices, missing values based on [6]). The hours per farm to collect the FLINT data (column 5) are the indications by the FADN committee members for their country. Based on these assumptions, the costs for adding the FLINT sustainability data were calculated (column 7 and 8).

**Table 1.** Costs of FADN and FLINT data collection in Europe (euros per farm).

| Country (1) | Type * (2) | Time per FADN Farm (3) Hours | Data Collection Cost per Hour (4) Euro per Hour | Time per Farm for FLINT Data (5) Hours | Current Costs FADN Farm (6) Euro(s) | Estimated Costs FADN Farm + FLINT Data (7) Euro(s) | (8) Increase in % |
|---|---|---|---|---|---|---|---|
| Austria | 2 | 16.8 | 46 | 10 | 1360 | 1819 | 34 |
| Belgium | 2 | 56 | 36 | 12 | 2000 | 2429 | 21 |
| Bulgaria | 2 | 15 | 14 | 10 | 209 | 348 | 67 |
| Croatia | 2 | 15 | 9 | 10 | 130 | 217 | 67 |
| Cyprus | 3 | 7 | 36 | 8 | 250 | 536 | 114 |
| Czech Republic | 2 | 30 | 12 | 9 | 370 | 481 | 30 |
| Denmark | 1 | 6 | 60 | 5 | 400 | 699 | 75 |
| Estonia | 2 | 28.8 | 14 | 9 | 314 | 439 | 40 |
| Finland | 2 | 25 | 40 | 7 | 1000 | 1280 | 28 |
| France | 3 | 4 | 52 | 12 | 500 | 1119 | 124 |
| Germany | 3 | 8 | 46 | 12 | 600 | 1157 | 93 |
| Greece | 3 | 24 | 53 | 12 | 1273 | 1910 | 50 |
| Hungary | 2 | 6 | 11 | 6 | 500 | 566 | 13 |
| Ireland | 1 | 24 | 42 | 2.5 | 1000 | 1104 | 10 |
| Italia | 1 | 15 | 20 | 8 | 300 | 460 | 53 |
| Latvia | 3 | 12 | 23 | 12 | 270 | 540 | 100 |
| Lithuania | 2 | 8 | 31 | 5 | 250 | 406 | 62 |
| Luxembourg | 2 | 50 | 40 | 15 | 2000 | 2600 | 30 |
| Malta | 2 | 4 | 25 | 9 | 100 | 325 | 225 |
| Netherlands | 1 | 54 | 56 | 6 | 3000 | 3333 | 11 |
| Poland | 2 | 32.2 | 20 | 10 | 656 | 860 | 31 |
| Portugal | 2 | 37.5 | 13 | 9 | 500 | 620 | 24 |
| Romania | 3 | 4 | 7 | 12 | 100 | 179 | 79 |
| Slovakia | 2 | 20 | 17 | 10 | 340 | 510 | 50 |
| Slovenia | 3 | 15 | 23 | 12 | 263 | 542 | 106 |
| Spain | 3 | 10 | 32 | 6 | 500 | 691 | 38 |
| Sweden | 2 | 9 | 56 | 12 | 800 | 1477 | 85 |
| UK ** | 2 | 44 | 45 | 9 | 2000 | 2409 | 20 |

* Type that classifies the member state on the current level of data collection on sustainability performance indicators and method for extra data collection. (1) comparable to Netherlands/Ireland: already many data available, FLINT data gathered in same process as FADN; (2) Comparable to Poland/Hungary: not many data available, FLINT data gathered in same process as FADN and (3) Comparable to France/Greece: not many data available, FLINT data gathered in a separate farm visit. ** EU includes the UK as the analysis was carried out and reported to the EU Commission before Brexit.

The estimated change in costs as displayed in column 8 in Table 1 shows a large range: from countries such as Ireland (+10%) and the Netherlands (+11%) to France (+124%) and Malta (+225%). Part of the explanation is that some of the countries already gather several data items of the FLINT data set for national purposes, but do not yet make them available to the EU FADN. Another is that some of the data-heavy FLINT farm return topics are not relevant in some countries, thus reducing the data collection burden (e.g., hardly any pesticides on crops in the Irish FADN sample where livestock dominates). Another is that countries with relatively low extra costs see options to integrate the data collection in the current process, where others are not able to do so or have to pay the full cost of such an adaption, as the current costs are relatively low since the FADN data are a by-product of tax accounting. This implies that the differences in costs between countries are lower in the desired situation where FADN data are supplemented by FLINT data than in the current situation with only FADN data: the burden is on those with relatively low costs in the current situation.

The results reflect a range of local circumstances. Based on their inventory, Bradley and Hill [6] concluded that the amounts differ due to differences in wage levels and due to differences in the scope of data collection. However, there are also other factors, such as whether the costs include direct labour costs only or a full commercial rate (including overhead costs and a profit margin); the quality of the data; special costs (e.g., the inclusion of costs of big ICT projects that once every ten or 15 years reorganise the software and working methods); and a potential under sourcing in some countries. These factors also affect the estimation of costs for the collection of FLINT data and confirm that data collection costs per hour and the required number of extra hours provide an acceptable basis for estimating the FLINT data collection costs.

*3.3. Estimation FSDN Costs at the EU Level*

On average (weighted with the number of farms per member state), the costs of adding the FLINT sustainability data to the FADN imply an increase from 750 euros per holding to 1040 euros. Adding the FLINT data to the full FADN sample of 85,000 farms would increase the costs by about 40%. Although these additional costs are low compared to the amount of subsidy payments and the pressing need for this information, available public resources are under scrutiny. To realise the FSDN, two options to reduce the costs of data collection without comprising the usability and statistical soundness of the results are explored.

In the first option, to maintain the costs at the same level at the country level, collecting the full set of FLINT sustainability performance indicators from all FADN farms would lead to a reduction of the FADN sample by a third, from 85,000 to 55,000 farms. Although this would not jeopardise the income estimation at the EU level, it would lead to considerable changes of the FADN panel in some countries such as France, Germany and Sweden. Such countries would most likely be confronted with unreliable estimates for some farm types at the regional level.

In the second option, FLINT data are only collected from a subsample of FADN farms. The calculations show that a sample size of 15,000 farms and an optimal allocation over countries would guarantee a relative standard error below 3%, and makes it possible to publish results for the most important farm types for the individual member states (see [19] for more details, see Supplementary Materials).

The sample allocation of these 15,000 farms, based on an optimal allocation over the member states, is given in Table 2. Combining that with the estimation of costs for data collection in Table 1 gives the reduction needed in the FADN sample per member state to collect these data within the current budget.

The calculations show that collecting the FLINT sustainability performance data from 15,000 farms would demand a reduction of less than 10,000 FADN farms, bringing the sample down from 85,000 to 75,000 farms. At the EU level, that is not a big loss in precision



of the income estimators. The FLINT sustainability performance data would then be gathered on 20% of the farms.

**Table 2.** Number of FADN farms per member state and with FLINT data collection for a subsample in the option FADN subsample (excluding Croatia, for which basis (2013) data were not yet available).

| Country | Current FADN Sample (Number of Farms) | Sample for FLINT Data (Number of Farms) | Increase in Cost (Fraction) | Required Reduction in FADN (Number of Farms) | Adjusted FADN Sample (Number of Farms) | % FLINT Farms |
|---|---|---|---|---|---|---|
| Belgium | 1228 | 360 | 0.06 | 77 | 1151 | 31 |
| Bulgaria | 2239 | 229 | 0.07 | 152 | 2087 | 11 |
| Cyprus | 469 | 23 | 0.06 | 26 | 443 | 5 |
| Czech Republic | 1401 | 274 | 0.06 | 82 | 1319 | 21 |
| Denmark | 1827 | 421 | 0.17 | 314 | 1513 | 28 |
| Germany | 8880 | 2089 | 0.22 | 1939 | 6941 | 30 |
| Greece | 4777 | 227 | 0.02 | 114 | 4663 | 5 |
| Spain | 8716 | 1907 | 0.08 | 729 | 7987 | 24 |
| Estonia | 660 | 41 | 0.02 | 16 | 644 | 6 |
| France | 7552 | 1946 | 0.32 | 2409 | 5143 | 38 |
| Hungary | 1972 | 380 | 0.03 | 50 | 1922 | 20 |
| Ireland | 938 | 150 | 0.02 | 16 | 922 | 16 |
| Italy | 10,929 | 3342 | 0.16 | 1782 | 9147 | 37 |
| Lithuania | 1067 | 50 | 0.03 | 31 | 1036 | 5 |
| Luxemburg | 444 | 9 | 0.01 | 3 | 441 | 2 |
| Latvia | 998 | 43 | 0.04 | 43 | 955 | 5 |
| Malta | 507 | 3 | 0.01 | 8 | 499 | 1 |
| Netherlands | 1513 | 899 | 0.07 | 100 | 1413 | 64 |
| Austria | 2119 | 161 | 0.03 | 54 | 2065 | 8 |
| Poland | 12,321 | 718 | 0.02 | 223 | 12,098 | 6 |
| Portugal | 2285 | 192 | 0.02 | 46 | 2239 | 9 |
| Rumania | 5881 | 430 | 0.06 | 340 | 5541 | 8 |
| Finland | 846 | 106 | 0.04 | 30 | 816 | 13 |
| Sweden | 1070 | 216 | 0.17 | 182 | 888 | 24 |
| Slovakia | 563 | 150 | 0.13 | 75 | 488 | 31 |
| Slovenia | 944 | 48 | 0.05 | 50 | 894 | 5 |
| UK | 2805 | 588 | 0.04 | 120 | 2685 | 22 |
| Total EU * | 84,951 | 15,000 | | 9011 | 75,940 | 20 |

* EU includes the UK as the analysis was carried out and reported to the EU Commission before Brexit.

## 4. Discussion

Given the Green Deal, the Farm to Fork Strategy and the broadening set of objectives for the Common Agricultural Policy, there is an increasing need for data on the sustainability performance of farms. The Farm to Fork Strategy proposes to develop the FADN into the FSDN to meet these needs. In this paper, we have estimated the additional costs of the FSDN.

Collecting FLINT data in national farm accounts surveys of the FADN will require adjustments in the systems. While we conclude that the costs of adaptation will be specific

to each member state, some general principles are evident. Although extending the collection of any type of data to the member states that do not currently collect it would incur costs, these would be marginal to the basic data collection infrastructure already in place. Innovation and learning processes might result in lower costs for the FSDN that we estimated in the FLINT project. Although (marginal) project costs are sometimes lower than the costs in a standing organisation, this is unlikely to be the case in our estimates. There are no learning and innovation effects involved in the estimate of the number of hours needed, and the price component was not based on the FLINT project costs but on the costs normally paid per hour in the standing FADN system.

Our calculations show large differences in FADN and FSDN cost per member state. This suggests that there is room to learn between member states on best practices [5] and that the development of the FADN into an FSDN could trigger innovations that can reduce our cost estimation.

Looking to the future, there are opportunities for further integration of sector and policy initiatives [35]. The costs of the FSDN (and its benefits) could be further increased by aligning the information needs of policy makers and those of the sector. Several industry schemes oblige farmers to collect sustainability data. The harmonisation between industry indicators and those used in FSDN and policy evaluation could reduce costs. Farmers will make data more easily available if data are already in their management software for the industry schemes. It reduces administrative burdens if farmers can supply the same data to the FSDN as to industry.

The FSDN objective is to provide quantitative information that helps policy makers make decisions or evaluate the impact of decisions for a country or farm type. There are many initiatives that measure sustainability performance in agricultural systems. The goal of the initiative determines what data should be assembled and which tools and indicators could be used to measure processes and practices. Despite the differences in goal and scope, there are opportunities for harmonisation and alignment between measurement frameworks, tools, and data assembling systems. At the product level, for example, The Sustainability Consortium (TSC) [36] convenes stakeholders in consumer good supply chains and develops science-based key performance indicators (KPI) that measure environmental and societal performance per product category based on a life cycle approach. Quantifying KPIs often requires farm-level data or regional estimates from a sub-country area or agricultural zone, which FSDN could provide.

Another question which heavily affects the costs (and benefits) of data collection is whether there is a need to collect all relevant data from the same set of farms, or whether the policy evaluation could be based on combining alternative data sources—and if this would be cheaper [13]. To answer this question, the FLINT project made a comparison between the situation where all data are collected from one farm and an estimation based on imputing data from other sources. For this purpose, a number of policy analyses were not only carried out with the integrated data collected in FLINT, but also with data that were imputed from other farms where FLINT data were collected (to mimic the situation that incomplete data are gathered on different farms and then combined). Results show that imputation often leads to degrading the explanatory power of the model and blurring the results regarding the relationships between the dependent variable and the chosen covariates. This strongly supports the line chosen for FSDN to collect sustainability indicators from the same set of farms.

Furthermore, policy makers have to evaluate the trade-offs between different policy objectives, e.g., farm income, different environmental impacts and food security (production levels). With policy measures, they try to influence the decision of a farmer in such a way that the outcome of the decision would be different from a situation without a policy. In policy evaluations, researchers try to compare these two situations: with and without a policy, in order to estimate the effectiveness of a policy. This asks for detailed data on the behaviour of the farmer and how their decision affects the policy objectives. It means that policy researchers are more interested in those relationships between policy,

management and the exact relation between inputs, outputs and income, than in the statistical data on use of inputs or of income as such. The necessity to consider the different farm level impacts was illustrated in the FLINT project [37]. The results show that the effect of subsidies on farms' technical efficiency changes when environmental outputs (i.e., greenhouse gas emissions, nitrogen balance and ecological focus areas) are taken into account in the efficiency calculation. Evaluations of policies that aim to improve efficiency should therefore be based on a full set of data in relation to the management decisions of the farmer.

Collecting sets of data items on different farms provides the advantage that the administrative burden of farmers could be spread if different farmers participate in different networks. However, it will most likely increase the total costs of data collection due to the duplication of costs items (such as farm visits, IT investments and development of (quality) procedures). The foregoing strongly advocates the gathering of data as exhaustively and precisely for the same farms and at the same time.

An argument brought forward against the FSDN is the fear that farmers will stop their participation if more data will be collected. Farmer participation depends on the balance between the burden and the value of participation. To ensure continued participation, the value should be increased by enabling the farmer to use the data to improve its farm performance (by providing sustainability reports, benchmarking, farm advisory services etc.) and to fulfil other information needs. On the other hand, the burden should be limited by making use of new technologies and digitalisation in the agricultural sector and exploiting the link between economic and environmental accounting. The collection of environmental data should not be an additional activity, but rather an integral part of the accounting process as already applied. Recognising the link between environmental and economic accounting also improves the quality of the data by crosschecking the financial and material flows.

Another concern raised is the claim that FADN is designed on economic criteria and therefore not suitable for environmental indicators [38,39], and that therefore our estimate of the number of farms to be represented in the panel could be wrong. As described by Vrolijk et al. [18], two aspects should be distinguished. These are the demarcation of the field of observation and the sample design of the FADN. The impact of the sample design is limited. Although SO (Standard Output) is defined as an economic indicator, also for collecting data on environmental and social issues, type of farming and size of farming would be important stratification variables. Due to the very strong correlation between physical size and economic size as measured by SO (especially within a type of farming) the resulting sample structure is likely to be very similar. The definition of the field of survey is not so much an economic issue but a policy decision. Farms smaller than the threshold are not included in FADN but do have an impact on the environment and the social dimensions of rural areas, especially in those regions with a large number of small or semi-subsistence farms. Here, it is important to be aware of the fact that FADN and FSDN are designed as tools to monitor and evaluate the Common Agricultural Policy. The Common Agricultural Policy is mainly aimed at commercial farms. Collecting sustainability data on FADN farms does not provide data on very small farms, but does provide the opportunity to evaluate the impacts of the CAP on economic, social and environmental objectives. If the CAP does focus on smaller farms, changing the field of observation of FADN should be considered, irrespective of whether or not sustainability data are collected.

The estimation of costs as presented in this paper can be regarded as an upper bound of the costs, as the efficiency of data collection can be improved by better re-use of already available data sources (administrative, commercial and statistical) and learning effects which increase the efficiency of data collection. In the event that the partners of the farmer in the food chain supplement or replace invoices on paper by digital versions, costs of management and financial accounting would drop substantially.

The FLINT project identified that many states use administrative data in compiling their FADN data set and how some have started to re-use commercial data that come from

invoices and other transaction data in a digital form. Costs of the FADN could be lowered substantially in the coming years if such a development were to take off, and farmers themselves would benefit most of all.

A European Innovation Partnership (EIP) Focus Group on Benchmarking [40] concluded that data sharing is an important theme for innovation. In the current situation, agri-businesses, such as sellers of farm inputs and buyers of farm produce, send tens of thousands of paper invoices and other documents per year to farmers. Farmers then have to type these data into their farm management information systems or accounting software. This is often restricted to the most needed data (e.g., financial data), where other data on the documents (on volumes of input and output or on quality indicators of the produce) are ignored, although these data would be useful for indicators on productivity and especially sustainability.

In the next years, this practice should evolve towards digital exchange with EDI (Electronic Data Interchange) messages. Novel more pro-active government approaches by public authorities could play a key role in promoting EDI approaches and benchmarking sustainability. The EIP Focus group [40] mentioned the block-chain technology as a possible solution, guaranteeing the ownership of data for the farmer and as such creating trust in a common interoperable system, which holds data that farmers may not want to share with all actors. Such principles as Single Entry and Digital by Default could help the agricultural sector and the food chain in managing its paperwork and administrative burden and increase the value and use of the data in benchmarking, research and policy evaluation.

## 5. Conclusions

The conclusion of the analysis is that collecting the sustainability data from all farms included in FADN to realise the Farm Sustainability Data Network would increase the costs from 750 to 1040 euros per farm, which is an increase by about 40% of the current cost of FADN. The results show large differences between countries depending on the current costs of data collection and the expected additional work to include sustainability indicators. The estimated change in costs ranges from 10% in countries such as Ireland and the Netherlands, to 124% in France, and even 225% in Malta.

A first version of the FSDN could be realised within the current FADN budget if the FLINT sustainability performance data were collected from 15,000 farms and the FADN sample were reduced by fewer than 10,000 FADN farms, bringing the sample down from 85,000 to 75,000 farms. At the EU level, that is not a big loss in precision of the income estimators, and for the main farm types, sustainability performance data would become available at the member state level. The FLINT sustainability performance data would then be gathered on 20% of the farms.

Given the pressing need for sustainability data and the conclusions on the costs of collecting sustainability data as reported in this paper, the rapid increase in the use of information technology in the farm sector and the increasing demands of the retail and food industry for farm sustainability, a viable organisational form for the collection of farm sustainability indicators can be developed. It is logical and a welcome development that the European Commission improves its monitoring and policy evaluation by extending the FADN into a Farm Sustainability Data Network.

**Supplementary Materials:** The following are available online at https://www.mdpi.com/article/10.3390/su13158181/s1, survey form, table with relative standard errors.

**Author Contributions:** Both authors contributed to all stages of the project. Both authors have read and agreed to the published version of the manuscript.

**Funding:** This paper is partly based on the FLINT project. This project has received funding from the European Union's Seventh Framework Programme for research, technological development and demonstration under grant agreement No. 613800.

**Institutional Review Board Statement:** Not applicable.

**Informed Consent Statement:** Not applicable.

**Data Availability Statement:** Restrictions apply to the availability of these data. FADN data was obtained from DG-Agri from the European Commission (see https://ec.europa.eu/info/food-farming-f isheries/farming/facts-and-figures/farms-farming-and-innovation/structures-and-economics/eco nomics/fadn_en) and access for research purposes can be requested from EU DG-Agri. The data on the cost of data collection obtained through the survey among FADN managers is published in this paper.

**Acknowledgments:** The authors thank the collaborating partners in the FLINT project, members of the FADN committee, researchers in the Pacioli network and participants of the 172nd EAAE Seminar "Agricultural policy for the environment or environmental policy for agriculture?", 28–29 May 2019 in Brussels for stimulating discussions.

**Conflicts of Interest:** The authors declare no conflict of interest.

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
