# Peer review of "Cost of Extending the Farm Accountancy Data Network to the Farm Sustainability Data Network: Empirical Evidence"

_sustainability, doi:10.3390/su13158181_

Round 1

Reviewer 1 Report

This is an interesting paper on the costs and benefits of farms, including a detailed analysis on the farm sustainability data network. Some specific points that could be improved are:

  • The objective of the paper should have to be introduced more clearly. In a way such as : "the main objective of this paper is...".
  • Definitions of the variables and sources should have to be more clearly stated. 
  • Sources of the data should have to be more clearly stated.
  • Some descriptive statistics for the different variables are needed (mean, median, standard deviation...). Some theoretical discussion on these variables could be interesting.
  • Location for the model should have to be more clearly explained.
  • The main conclussions of the model, (comparing with similar models), should be very useful.

Reviewer 2 Report

The paper targets a relevant topic by addressing the benefits and economic challenges of sustainability data collection at farm level, and is therefore in principle worthy of publication.  However, there are certain issues with the manuscript that limit its contribution. In the following, I highlight my most important concerns.

General comments:

Overall, the paper is lacking a stringent structure, making it difficult to follow what the authors are presenting.

In the introduction, it is sometimes not clear to me whether statements are the authors' ideas or those from other sources (e.g. line 40, 47, 53, 80, 101….).  The same applies to the discussion section (e.g. line 345, 360-370, …) Therefore, the authors should please be more careful in citing other research.

They should also be careful to clearly define abbreviations by putting them in parentheses after the spelled out version before using them in the rest of the manuscript (e.g., CAP, FADN, FLINT)

Introduction:

In the introduction, the authors do not elaborate on the issue of FADN (or comparable) data collection costs and the importance of these data collection costs, e.g., the importance to stakeholder acceptance of such data collection.

The authors write: "Given these conclusions the ultimate question is whether this pilot should be prolonged and extended to all EU member states" Is this the research question of the paper? If yes, then this question should be referred to in the discussion and a clear answer to this question should be given.

Methods:

In the methods section it is not clear to me if the authors have collected own data or if and which data are from other research/sources.

The authors write that a "typology of organizational systems was used to categorize the countries." Please explain the typology in detail, including the characteristics of the categories.

The methods section must describe the process of data collection and processing in such a way that other researchers could theoretically replicate the research. However, the methods section would not allow this in its current version.

Results:

The results section contains text that would belong in either the introduction, the methods section, or the discussion. So please be clearer and more consistent in assigning passages of text to their respective sections.

3.1 For the benefits, it would be helpful to create a table, e.g., with one column naming each benefit, a second with a brief explanation of why it is a benefit, and a third with the source(s).

3.2.2 It is not entirely clear to me how the authors estimated the FADN and FADN + FLINT costs and how outliers were defined (this should be described in the methods section).

Discussion:

In the discussion section, the authors refer to focus group discussions (line 450 ff) What focus group discussions? There is nothing written about this in the methods and results section. Or is a citation missing in this paragraph?

Reviewer 3 Report

Summary

The paper discusses the costs of collecting farms sustainability performance data for the European Common Agricultural Policy program designed to mitigate environmental and climate impacts.

Major comments

The study attempts to estimated the costs of collecting sustainability performance data and one of the key inputs is the number of hours the data collectors spend collecting data from individual farmers. I believe this approach would tremendously underestimate the costs if the data collectors are employed on the project rather than being paid on an hourly basis. The authors are encouraged to clarify this aspect in their methodology. 

The section on benefits of data collection is weak. It is a mere description of the item that could be considered as part of the benefits quantification. I believe it does not really reflect the concept of "cost-benefit analysis" in economics. Therefore, I would suggest the authors change the title of the manuscript and remove the benefits aspect. The benefits section could be part of the Discussion section. In addition, I am wondering whether the "identification of the usage of the data" is not too narrow a factor to account for the benefits of the data. I believe a more in-depth analysis in terms of, for example, actionable insights/corrective measures that the data allow for could strengthen the paper. 

Minor comments

L11: evidence-based

L19: There is a semantic issue needs to be resolved throughout the paper. I suggest to change the term "sustainability data" to "sustainability performance data". 

L27: Green Deal

L38: idem L11

L39: typo: extend 

L60: stakeholders

L70: the word "farms" is repeated. The second one can be removed.

L125: I suggest to change the narrative from "data collection on (one) farm(s)" to "data collection from one farmer or farmers" throughout the paper. Therewith, the authors will avoid the confusion that the data collectors had to deploy their own sensors on farms to monitor sustainability performance. 

L165: A grammatical error needs to be corrected: it should be either "the analysis was" or "the analyses were". 

L217: Update the caption of the figure and add "Average Time ..."

L246-250: This portion of the paragraphs seems more appropriate for the Materials and Methods section. 

Reviewer 4 Report

The abbreviations i.e. FADN, FSDN, FLINT can make confusion to readers, esspecially for first time readers. I can understand, that authors are familiar with them, but the papers are published to wide public and the intension of MDPI papers is to make the topic presented in the paper clear from the begining. There is no Conclusions in the paper at all, and should be introduced.

Figures should be presented using bigger letters.

Table 1, column 6, Current costs FADN farm - no currency

Table 1, If there are not much data available in some countries, so such countries should not be considered in evaluation.

General comment for MDPI papers structure:

Title

Abstract

Key words

Abbreviations and symbols

1. Introduction

2. Materials and methods

3. Results

4. Discussion

5, Conclusions

6. References

Round 2

Reviewer 1 Report

The definition of the concept "FADN", should have to be found in the presentation of the paper, (in the title), and not only in Introduction.

Reviewer 2 Report

The paper has been considerably improved in terms of content, clarity and stringency. It can be published in its present form.

Reviewer 3 Report

Minor comments

Table 1: minor changes must be made to two of the columns. "Hours per ..." should be "Time per ...". In addition, the header of column 7 and 8 must be formatted as the other columns, i.e., units are placed below the column numbers. 

Euro(s) must start with capital letter. 

L375: fulfill instead of fulfil.

L414: The acronym EIP is not defined. 
